# Assessing Amylose Content with Iodine and Con A Methods, In Vivo Digestion Profile, and Thermal Properties of Amylosucrase-Treated Waxy Corn Starch

**DOI:** 10.3390/foods13081203

**Published:** 2024-04-15

**Authors:** Inmyoung Park, Mohamed Mannaa

**Affiliations:** 1School of Food and Culinary Arts, Youngsan University, Busan 48015, Republic of Korea; 2Department of Integrated Biological Science, Pusan National University, Busan 46241, Republic of Korea; mannaa@cu.edu.eg; 3Department of Plant Pathology, Faculty of Agriculture, Cairo University, Giza 12613, Egypt

**Keywords:** amylosucrase, starch modification, amylose content, in vivo digestion, thermal properties, low glycemic starch

## Abstract

In this study, waxy corn starch was modified with 230 U or 460 U of amylosucrase (AS) from *Neisseria polysaccharea* (NP) to elongate the glucan. The amylose content of the AS-modified starches was determined using iodine and concanavalin A (Con A) methods, and their in vivo digestion, thermal, swelling, and pasting properties were evaluated. The amylose content of AS-treated starches was not significantly different (*p* > 0.05) when using the Con A method but was significantly higher than that of non-AS-treated samples when using the iodine method. In vivo, rats fed AS-treated starch had significantly lower blood glucose levels at 15 min than other rats; rats fed 460 U AS had lower blood glucose levels at 30 and 60 min than non-AS-treated rats. DSC analysis revealed that AS-treated starches exhibited higher initial, melting, and completion temperatures. Minimal volume expansion was observed by swelling factor analysis, while a Rapid Visco Analyzer assessment revealed that they had higher pasting onset temperatures, lower peak viscosities, and no trough viscosity compared to native starch. The elongated glucans in AS-treated starch reinforced their crystalline structure and increased slowly digestible and enzyme-resistant starch content. Overall, AS-treated starch showed unique thermal properties and a reduced blood glucose index upon administration. This distinctive characteristic of NPAS-treated starch makes it a good candidate food or non-food material for cosmetic products, medical materials, and adhesives.

## 1. Introduction

Starch is the major dietary source of carbohydrates in humans, playing an important role in energy supply. Based on their rate of glucose release and absorption in the intestine, starches can be classified into rapidly digestible (RDS, digested within 20 min), slowly digestible (SDS, digested between 20 and 120 min), or enzyme-resistant starch (RS, not digested in the gastrointestinal tract) [1]. SDS undergoes a slow digestion process until its complete degradation in the small intestine; therefore, it is the main determinant of the postprandial glucose level. RS escapes digestion in the small intestine but is fermented in the large intestine by colonizing bacteria, producing short-chain fatty acids that also act as prebiotics, resulting in increased bacterial mass [2,3]. Both SDS and RS have a low glycemic index (GI); therefore, they have the advantage of reducing the risk of developing diabetes, obesity, and cardiovascular diseases [1,4,5]. Moreover, they are widely used as raw materials to reduce the GI of food produced using various flour products, such as noodles, bread, and biscuits [6,7,8].

Amylosucrase from *Neisseria polysaccharea* (NPAS) is a remarkable glucosyltransferase (EC 2.4.1.4) that elongates glucose chains by synthesizing linear α-(1,4) glucans using sucrose as a substrate in the absence of any primer, synthesizing amylose-like polymers in vitro [9,10,11]. de Montalk et al. [10] reported the production of the degree of polymerization (DP) of 55 oligosaccharides in vitro with NPAS using sucrose as a substrate. NPAS was then applied to various types of starch, such as waxy, normal, and high-amylose wheat, corn, and potato starches, where the non-reducing end of the amylose and amylopectin (AP) external chains were elongated by 13–23 glucose units by creating α-(1,4) glycosidic bonds [12,13,14,15,16].

After elucidating the molecular structure of NPAS using various analytical tools, when elongating the external branch chains of AP, the proportion of the shorter A and B1 chains decreased, while the proportion of the longer B2 and B3 chains increased [12,13,14,15,16]. An abundance of long AP side chains tends to generate longer double helices (helices with more turns), which generates more stable double helices and strengthens the crystalline region [17,18]. Therefore, NPAS-treated starches are less susceptible to hydrolytic enzymes due to their increased SDS and RS content [13,14,15].

From a practical point of view, the functional properties of starch, such as gelatinization, retrogradation, swelling behavior, and pasting properties, are considered critical for understanding their molecular behavior during the heating and cooling cycle. This, in turn, is related to their cooking characteristics and potential industrial applications. When starch granules are heated with water, they swell irreversibly and their crystalline structure collapses in a process known as gelatinization [19,20]. The swelling behavior of cereal starch is dependent on the proportion and branch chain properties of AP, while amylose acts as its inhibitor. Therefore, the ratio of amylose to AP and the detailed crystalline structure of AP affects the swelling of starch granules, consequently affecting their functional (thermal) properties [8,20,21,22]. NPAS-treated starch is known to contain altered elongated AP branch chains and an altered crystalline structure; hence, it would be valuable to evaluate the thermal properties of NPAS-treated starches to understand their cooking behavior and determine their food and industrial applications.

Therefore, this study aimed to observe and characterize the amylose content, in vivo digestion profiles, and thermal properties (gelatinization behavior, swelling pattern, and pasting properties) of starch prepared using two different NPAS doses (230 U and 460 U) and compare them with those of native and control starch to determine their cooking behavior and potential industrial applications.

## 2. Materials and Methods

### 2.1. Materials

Waxy corn starch was donated by MSC Corp. (Yangsan-si, Kyungsangnam-do, Republic of Korea). Sucrose, pancreatin (P7545, activity 8 × USP/g), and amyloglucosidase (AMG 50 mL, activity 300 AGU/mL) were obtained from Sigma Chemical Co. (St. Louis, MO, USA). Isoamylase was purchased from Megazyme (E-ISAMY, Wicklow, Ireland). All chemicals were of at least analytical grade.

### 2.2. NPAS and Preparation of AS-Treated Starch

NPAS was kindly provided by the Food Microbiology and Biotechnology Laboratory of Kyunghee University (Suwon-si, Kyunggi-do, Republic of Korea). The gene of NPAS was cloned and expressed in *Escherichia coli*, and NPAS was purified using affinity chromatography with Ni-NTA (nickel-nitrilotriacetic acid: Qiagen, Hombrechtikon, Switzerland). AS activity was determined according to Jung et al. [23]. Enzyme-modified starch was prepared as described in a previous study [13]. Briefly, a 2% (*w*/*w*) starch suspension was prepared by dissolving starch in a 100 mM sodium citrate buffer (pH 7.5). To this suspension, 100 mM sucrose was added as a substrate, followed by boiling for 30 min. After cooling to 30 °C, each NPAS (230 U and 460 U) was introduced into the mixture and then incubated at 30 °C for 20 h in a water bath, and then the reaction was terminated by adding ethanol (×3 volume). Subsequently, the AS-treated starch was separated by centrifugation (10,000× *g*, 10 min). The residual precipitate was washed with distilled water, followed by centrifugation (10,000× *g*, 10 min) to eliminate soluble components; this step was repeated 3 times. The final pellet was freeze-dried, pulverized, and sieved through a 100-μm mesh. For control samples, starch was processed identically, excluding the addition of AS.

### 2.3. Amylose Content: Iodine Colorimetry and Concanavalin (Con A) Method 

Apparent amylose content was measured by iodine colorimetry according to Jane et al. [24], with slight modifications. Purified starch was accurately weighed and heated in 4 mL of 90% DMSO in a boiling water bath for 15 min. After that, 200 μL of this solution was diluted with 8.8 mL of distilled water, followed by the addition of 1 mL of lugol solution (Duksan Pure Chem., Ansan-si, Kyunggi-do, Republic of Korea). After 15 min at room temperature, the absorbance of the sample was read with a spectrophotometer at 660 nm. A standard curve was generated with 0, 2.5, 5, 10, 20, and 30% amylose solutions using a mixture of amylose from potato (Sigma Chemical Co.) and starch (waxy corn, MSC Corp.). Alternatively, amylose content was determined using an Amylose/AP assay kit (K-AMYL, Megazyme) based on the Con A-lectin method [25]. Con A formed complexes with AP (branched polymer) and was then precipitated, but not with linear amylose. After removing the AP-Con A complex, the amylose broke down to D-glucose after being treated with α-amylase and amyloglucosidase. The D-glucose was then measured calorimetrically after treatment with a glucose oxidase and peroxidase reagent (GOPOD). The concentration of amylose in the starch sample was estimated as the ratio of GOPOD absorbance at 510 nm of the supernatant of the Con A precipitated sample to that of the total starch sample. The detailed procedure followed the manufacturer’s protocol.

### 2.4. In Vivo Digestibility Using the Rat Model

Spraque–Dawley male rats (8 weeks old, 250 ± 20 g) were individually housed in an approved laboratory animal facility for a 7-day adaptation period. The rats fasted for 16 h and were then orally administered 500 µL sample suspensions (7.5%, *w*/*v*) or glucose (7.5%) as treatments. Blood samples were taken twice each time from the tail vein of each rat at 0, 15, 30, 60, 120, and 240 min, respectively. Blood serum glucose levels were measured instantly with an Accu-Check Active Performer (Roche Ltd., Basel, Switzerland). The in vivo digestibility was measured using seven replicates. The IACUC approved the animal experiments (approval number: QBSIACUCA16107).

### 2.5. Measurement of Thermal Properties

#### 2.5.1. Differential Scanning Calorimetry (DSC)

Thermal properties of the starches were investigated using the differential scanning calorimeter DSC8000 (Perkin-Elmer, Waltham, MA, USA) equipped with an intracooler 2 and Pyris Series. Starch samples (3 mg each, db) were weighed into aluminum sample pans (#0219-0062, Perkin-Elmer), mixed with distilled water (50% *w*/*w*), sealed, and allowed to reach equilibrium for 4 h at room temperature. The instrument was calibrated with indium, and an empty pan was used as a reference pan. Heating was performed at a temperature range of 30–120 °C at a rate of 10 °C/min. Onset (T_o_), peak (T_p_), conclusion (T_c_), gelatinization temperature, gelatinization temperature range (T_c_–T_o_), and gelatinization enthalpy (ΔH, J/g) were determined using Phyris software, version 3.81 (Perkin-Elmer).

#### 2.5.2. Swelling Factor (SF)

The SF of each starch was determined using the method of Tester and Morrison [19]. Starches (50 mg, db) in water (5 mL) were incubated with constant shaking in a water bath at 50, 65, 80, and 95 °C for 30 min, respectively. The samples were then cooled to 25 °C for 15 min, followed by the addition of 0.5 mL of blue dextran (Sigma, Co., Cream Ridge, NJ, USA, M_W_ 2 × 10^6^, 5 mg/mL), and then the tubes were gently inverted to mix the contents. After centrifugation at 2000× *g* for 5 min, the absorbance of the supernatant was measured at 620 nm. The SF was calculated using the equation [19]:SF = 1 + [(7700/W)[(A_S_ − A_R_)/A_S_]]
where W represents the sample weight, A_R_ represents the absorbance of the supernatant, and A_S_ represents the absorbance of the sample.

#### 2.5.3. Pasting Properties

The pasting properties of each starch (10% db) were measured using a Rapid Visco Analyzer (RVA-4, Newport Scientific Pty Ltd., Warriewood, Australia). Briefly, samples were equilibrated at 50 °C for 1 min, heated at 12 °C/min to 95 °C, held at 95 °C for 2.5 min, cooled at 12 °C/min to 50 °C, and held at 50 °C for 2 min. Pasting temperature, peak viscosity, setback, breakdown, and final viscosity were determined to explain the pasting properties.

### 2.6. Statistical Analysis

The data are reported as the means of triplicate measurements with standard deviations. Analysis of variance (ANOVA) was performed and differences in means of samples were analyzed using Duncan’s multiple range tests (*p* < 0.05) with SAS software (version 9.3., SAS, Cary, NC, USA). For statistical analysis of in vivo digestibility, ANOVA and Dunnett’s multiple comparison tests were carried out using GraphPad PRISM software (version, 5.0, GraphPad Software, San Diego, CA, USA).

## 3. Results and Discussion

### 3.1. Determination of Amylose Content Using the Iodine and Con A Methods

NPAS-treated waxy corn starch contains elongated glucose molecules with extended α-(1,4) linkages at the non-reducing end of the AP branch chains. These elongated branch chains are much longer than those of the original AP; therefore, they have “amylose-like” properties.

To investigate the properties of the elongated glucans in NPAS, the amylose contents of native, control, and AS-treated (230 U and 460 U) waxy corn starches were determined using the iodine and lectin Con A methods (Table 1). Using the iodine method, the amylose content was significantly higher in both 230 U (9.79%) and 460 U AS-treated starches (12.89%) than in native (2.01%) and control starches (0.32%) (*p* < 0.05). Using the Con A method, the amylose content was determined to be 1.37% and 1.35% in 230 U and 460 U AS-treated starches, respectively, which were not significantly different (*p* > 0.05) compared with the 1.21% and 1.15% in native and control starches, respectively. 

AS treatment overestimates the amylose content because it extends the linear α-(1,4) glycosidic bonds at the ends of the AP side chains. The newly extended glucan polymer then behaves similarly to an amylose molecule. Iodine molecules could bind inside the helix of this newly created glucan polymer [26], leading to further blue staining and overestimating the amylose content. In contrast, the Con A method only measures the amylose content after removing the AP-Con A complexes. Therefore, amylose content measured using the Con A method is not affected by the AP side chain length whether it is shorter or longer. Using the Con A method revealed that AP, with its elongated branched chains produced by AS treatment, exhibited behaviors similar to those of long linear amylose and amylose-like molecules [15,27]. Although the distribution of specific branch lengths of NPAS-treated starch is usually determined using high-performance anion-exchange chromatography [12,13,15,28], comparing the results of the iodine and Con A methods might also preliminarily identify the side chain length properties of NPAS-treated starches.

### 3.2. In Vivo Digestibility Assay Using a Rat Model

The postprandial blood glucose response profiles of rats after administering waxy corn starch, control starch, and 230 U and 460 U AS-treated waxy corn starches are shown in Figure 1. Glucose was used as a positive control. Glucose and waxy corn starch (gelatinized starch) sharply increased the postprandial blood glucose levels 15 and 30 min post-administration. Afterward, the postprandial glucose levels dropped steeply at 60 and 120 min. These samples seemed to be very easily digested and absorbed, rapidly increasing the blood glucose levels. Native starch showed a moderate increase in blood glucose levels at 15 and 30 min, lower than that induced by glucose and waxy corn starch and higher than that induced by AS-treated starches; the blood glucose level decreased after 30 min. The postprandial blood glucose level of the rats administered the two AS-treated starches was significantly lower at 15 and 30 min (*p* < 0.05) than that of rats treated with glucose. However, the administration of 460 U AS-treated starches resulted in higher blood glucose levels than glucose at 120 min (Figure 1).

These results suggest that the AS-treated waxy corn starches contained a substantial amount of SDS, which continuously supplies blood glucose during digestion at a slower pace than the RDS fraction. While both AS-treated starches induced the lowest glucose levels after 15–30 min, the glucose level of rats administered 230 U AS-treated starch was not significantly different from that of the other rats after 60 min. Meanwhile, rats administered 460 U AS-treated starch had the lowest postprandial blood glucose levels at 30 and 60 min, which was significantly different from that of rats administered other samples (*p* < 0.05). However, it was significantly higher than that of glucose-administered rats at 120 min (*p* < 0.05). 

RDS is digested in the gastrointestinal tract within the first 20 min of ingestion, while SDS is digested between 20 and 120 min in the small intestine. The current results showed that 230 U AS-treated starch is more effective in continuously supplying glucose to the bloodstream at a slower pace during digestion than native starch. Moreover, 460 U AS-treated starch released a much lower amount of glucose than the other samples, which could explain the observed blood glucose level at 20–120 min post-administration (Figure 1). From these data, it can be postulated that 230 U AS-treated starch might contain a larger proportion of SDS than the other starch samples, while the 460 U AS-treated starch contained a smaller SDS fraction and larger RS fraction than the other samples. This postulation agrees with previous results of starch digestibility in vitro assay, where control starch showed an 85.3% RDS fraction, 230 U AS-treated waxy starch contained 30.3% SDS and 9.2% RS, and 460 U AS-treated starch contained 42.7% SDS and 32.7% RS [8,13]. This previous report demonstrated that the AS dose affects the SDS and RS fractions in the modified starch. Overall, the digestion patterns of AS-treated starches from these in vitro assays using digestion enzymes and our in vivo experiments using rat models were consistent. Moreover, the elongation of the external branch chains of AP, the formation of double helices, intercluster connections, crystallinity, and branch chains favor retrogradation, thereby hindering enzyme access and decreasing the rate of enzyme digestion [15,24,29,30,31].

### 3.3. Thermal Properties of the Investigated Starches 

The thermal properties of native waxy corn, control, and AS-treated starches were examined using DSC, SF, and RVA.

#### 3.3.1. DSC Studies

The onset (T_o_), peak (T_p_), and conclusion (T_c_) temperatures; melting temperature range (T_c_–T_o_); and melting enthalpies (ΔH) of the native, control, and NPAS-treated waxy corn starches are described in Table 2. Native waxy corn starches showed typical DSC endothermic peaks for gelatinization (T_o_ = 66.8 °C, T_p_ = 70.3 °C, T_c_ = 77.5 °C, T_c_–T_o_ = 10.7 °C, and ΔH = 15.0 J/g), whereas the control starch did not show any endothermic peak because it was already gelatinized. The endothermic peaks of NPAS-treated starches (both 230 U and 460 U) revealed a broader shape, with higher T_o_ (80.8 and 74.6 °C, respectively), T_p_ (94.9 and 87.9 °C, respectively), and T_c_ (100.4 and 100.4 °C, respectively); a wider range of gelatinization temperatures (T_c_–T_o_ = 19.6 and 25.8 °C, respectively); and lower gelatinized enthalpies (ΔH = 6.5 and 7.9 J/g, respectively) than those of native starch. These results indicate that the thermal characteristics of the modified starches were elevated.

Waxy corn starch contains shorter A chains of AP, which are too short to form stable helices. Thus, it would require less energy to unravel and melt during gelatinization than NPAS-treated starch. In comparison, the elongated AP side chains in NPAS-treated starch exhibit increased AP–AP interactions, resulting from an increased degree of crystallite heterogeneity within the granules. Thus, NPAS-treated waxy corn starch could form more densely packed double helices and crystallites than native starch, resulting in increased T_o_, T_p_, and T_c_ values, consequently increasing the T_c_–T_o_, which is a good agreement with previous studies [32,33,34]. Previous studies also reported that the increase in T_p_ and T_c_ suggests that long AP chains formed long-range double helices and more densely packed double helices, increasing their thermostability [13,18,29,35].

Furthermore, 460 U AS-treated starch had lower T_o_ and T_p_ values, a wider T_c_–T_o_, and higher ΔH than the 230 U AS-treated starch (Table 2). This result indicates that 230 U AS-treated starch has a higher starting temperature of gelatinization (T_o_) and a higher temperature to reach maximum gelatinization (T_p_) than 460 U AS-treated starch. However, it has a lower ∆H than 460 U AS-treated starch; ∆H shows the degree of molecular order loss inside starch granules [19]. As the amount of AS added varies, the temperature rise of the thermal properties of NPAS-treated starch tends to be similar to the results of previous studies [14,15,27]. Results using *Deinococcus geothermalis*-derived amylosucrase-modified potato starches showed that a high proportion of DP (≥25) induced high T_o_, T_p_, and T_c_ values, while no effect was observed for ΔH [28], which corresponds with the results of our current study. 

#### 3.3.2. SF

The SF values of the starches at 50–95 °C are presented in Figure 2. The SF of native waxy corn starch increased from 8.81 to 20.23 as the temperature increased from 50 °C to 65 °C. It more steeply increased between 65 °C and 80 °C, finally reaching an SF of 66.5 at 95 °C. The control starch had an SF of 58.7 at 50 °C and reached a maximum of 70.0 at 95 °C. From these observations, an SF of 70 seems to be the maximum water sorption power of these starches. This finding indicates that the control starch is very easily swollen by water, even at an initial temperature of 50 °C. It might lose its crystalline region during the boiling step and exhibit retrogradation during the cooling step as a native starch. Control starch mostly comprises amorphous regions; the resulting amorphous structure allows it to reabsorb water quickly. In contrast, AS-treated starches were observed to be very resistant to swelling. Their SF only increased slightly as temperature increased, from 3.0 at 50 °C to 8.0 at 95 °C in 230 U AS-treated starch and from 6.4 at 50 °C to 10.3 at 95 °C in 460 U AS-treated starch (Figure 2).

The SF is also affected by the elongated AP branch chains in AS-treated starches; the long branch chains of AP may contribute to a stronger crystalline structure, suppressing swelling [18,26,28,31]. Additionally, the elongation of external glucose molecules reduces the number of short A (DP 6–12) chains in AP, resulting in a decreased proportion of the amorphous region, an increased proportion of the crystalline region, and increased crystalline stability [16,28]. A previous study reported that native starch contained 22.4% A chains, but 230 U and 460 U NPAS-treated starches only contained 7.1 and 5.7% A chains, respectively [13].

#### 3.3.3. Pasting Properties

The pasting properties of the starches were measured using RVA to observe the pasting onset temperature, peak, final, trough, breakdown (final–trough), and setback viscosity (final–peak). As shown in Figure 3, the control and NPAS-treated starches showed unusual pasting viscosity profiles under programmed heating–cooling conditions. Native waxy corn starch displayed a typical profile, with a pasting onset temperature of 73.3 °C, peak viscosity of 23.4 RVU, trough viscosity of 114 RVU, and a final viscosity of 131 RVU. The control starch had no detectable pasting onset temperature and lower peak, breakdown, or setback viscosity compared to native starch; this is because the control starch was already gelatinized, and its crystalline structure was previously disrupted. Incubation at 30 °C in a water bath for 20 h resulted in a slight retrogradation of the gelatinized starch molecules and a partial re-association of its crystalline structure. As a result, no pasting onset temperature was observed, and the partial retrogradation resulted in lower peak, trough, and final viscosities.

AS-treated starches showed higher pasting onset temperatures than native and control starches. The pasting onset temperature of 230 U and 460 U AS-treated starch substantially increased, reaching 79.5 °C and 86.6 °C, respectively. The peak viscosity and time of 230 U and 460 U AS-treated starches were determined to be 93 RVU at 7.8 min and 121 RVU at 8.1 min, showing a peak viscosity shift to the right, corresponding to the time of trough viscosity of native and control waxy starch. At a programmed holding temperature of 95 °C with shear stress, a starch sample is further destroyed, and the amylose molecules leach out of the solution, resulting in decreased viscosity. The viscosity at this time is called the trough viscosity. However, the AS-treated starch uniquely delayed the granule expansion and reached the trough viscosity later than the other samples, and when looking at the RVA profile of AS-treated starch, we cannot be sure that the destruction of the starch granules was completed (Figure 3). Consequently, the final viscosity defined as the viscosity at all RVA programs ends after a cycle of heating at 50 °C to 90 °C, holding at 95 °C, and cooling again to 50 °C. However, the final viscosity of AS-treated starch differed in terms of molecularity from the originally defined final viscosity because it is rarely applied to AS-treated starch and cannot complete degradation and recombination of starch molecules during programmed times as those used for AS-treated starch. Overall, the pasting properties of enzyme-treated starch samples showed a long initial expansion time, delayed paste onset temperature, and decreased peak and final viscosity, indicating crystalline stability. Failure to determine trough viscosity upon degradation indicates that AS-treated starch particles did not reach their maximum expansion volume or that the granules did not complete their destruction of starch molecules. 

Native corn starch showed conventional gelatinization and pasting properties, as in a previous study [14]. NPAS-treated waxy starch showed an increased pasting temperature and peak viscosity and much higher storage (G′) and loss (G″) moduli than native starch, indicating the enhancement of gel strength following AS treatment [14,15]. This observation is related to the elongation of AP, as a previous study reported that AP with long branch chains showed retarded starch swelling and an inclination to form a gel [19,22,31].

Peak viscosity is often correlated with final product quality, providing an indication of the viscous load during processing. The pasting profile of AS-treated starches also showed a unique pattern; thus, the results of this study suggest that AS-treated starches can be used not only as a low-GI foodstuff but also as a new viscous material for industrial food applications.

## 4. Conclusions

In summary, the present study investigated the impact of AS treatment on the physicochemical properties of starch. The amylose content of the NPAS-treated starches was consistently lower than that of the control, as confirmed using iodine colorimetry and the Con A method. The in vivo digestion profiles revealed that NPAS-treated starches exhibited lower susceptibility to enzymatic digestion, leading to reduced glucose release, than the control. Thermal analysis indicated alterations in the gelatinization behavior of NPAS-treated starches, showing slightly higher onset temperatures and lower gelatinization enthalpies than the native and control starches. SF measurements demonstrated changes in the water absorption and dissolution properties of NPAS-treated starches. Pasting property analysis revealed variations in the cooking and cooling behavior of NPAS-treated starches, suggesting potential applications in the food industry where modified starch functionality is desired. In future research, it will be meaningful to adjust the RVA program to measure the through viscosity and final viscosity of AS-treated samples, which is starch that does not easily swell and has not been studied for molecular re-association during retrogradation. Overall, the findings contribute to understanding the effects of AS treatment on starch properties, providing valuable information for developing starch-based ingredients with tailored functionalities.

## Figures and Tables

**Figure 1 foods-13-01203-f001:**
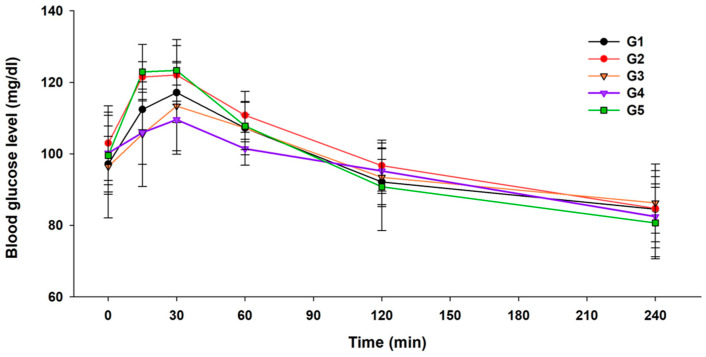
Mean blood glucose level in rats after the intake of native waxy corn starch (G1), control starch (G2), 230 U amylosucrase (AS)-treated starch (G3), 460 U AS-treated starch (G4), or glucose (G5).

**Figure 2 foods-13-01203-f002:**
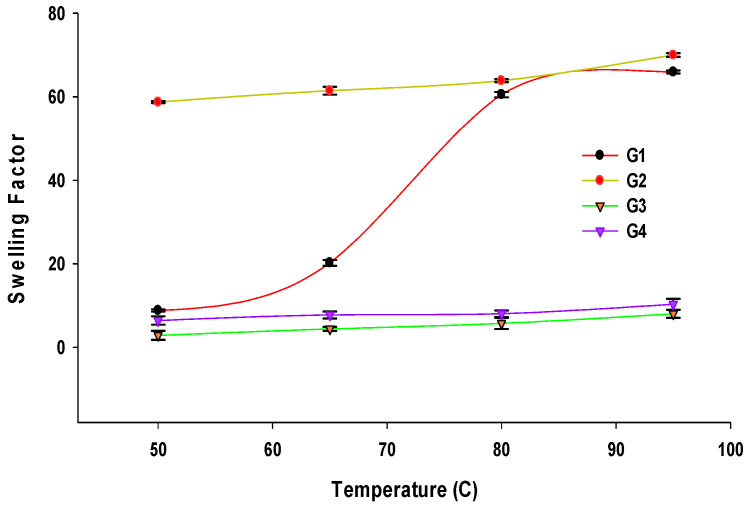
Swelling factors of native waxy corn starch (G1), control starch (G2), 230 U amylosucrase (AS)-treated starch (G3), and 460 U AS-treated starch (G4).

**Figure 3 foods-13-01203-f003:**
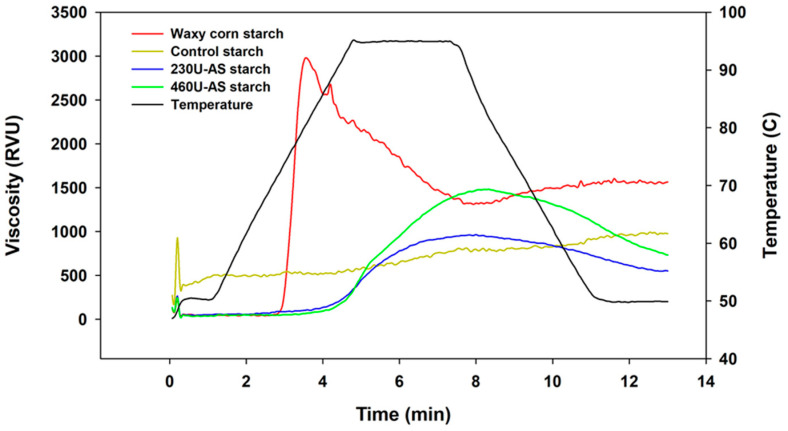
Pasting properties of native waxy corn starch, control starch, 230 U amylosucrase (AS)-treated starch, and 460 U AS-treated starch.

**Table 1 foods-13-01203-t001:** Amylose content of native waxy corn starch, control starch, 230 U amylosucrase (AS)-treated starch, and 460 U AS-treated starch using two different methods.

	Amylose Content (%)	
Sample	Iodine Method	Con A Method
Native	2.01 ± 0.0 ^c^	1.21 ± 0.1 ^a^
AS-Con	0.32 ± 0.1 ^d^	1.15 ± 0.3 ^a^
230U	9.78 ± 0.3 ^b^	1.38 ± 0.3 ^a^
460U	12.89 ± 0.2 ^a^	1.35 ± 0.2 ^a^

The values with different superscripts in each row are significantly different (*p* < 0.05) by Duncan’s multiple range test.

**Table 2 foods-13-01203-t002:** Gelatinization parameters of native waxy corn starch, control starch, 230 U amylosucrase (AS)-treated starch, and 460 U AS-treated starch.

Sample	T_o_ (°C) ^(1)^	T_p_ (°C)	T_c_ (°C)	T_c_–T_o_ (°C)	ΔH (J/g)
Native	66.8 ± 0.6 ^c^	70.3 ± 0.5 ^c^	77.5 ± 1.4 ^b^	10.7 ^c^	15.0 ± 1.7 ^a^
230U	80.8 ± 0.5 ^a^	94.9 ± 0.5 ^a^	100.4 ± 0.9 ^a^	19.6 ^b^	6.5 ± 1.0 ^b^
460U	74.6 ± 3.0 ^b^	87.9 ± 1.4 ^b^	100.4 ± 3.2 ^a^	25.8 ^a^	7.9 ± 1.4 ^b^

The value with different superscripts in each row is significantly different (*p* < 0.05) by Duncan’s multiple range test. ^(1)^ T_o_: onset melting temperature, T_p_: peak melting temperature, T_c_: conclusion temperatures, T_c_–T_o_: melting temperature range, ΔH: melting enthalpies.

## Data Availability

Data is contained within the article.

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
