# Peer review of "Assessing Amylose Content with Iodine and Con A Methods, In Vivo Digestion Profile, and Thermal Properties of Amylosucrase-Treated Waxy Corn Starch"

_foods, 2024, doi:10.3390/foods13081203_

Round 1

Reviewer 1 Report

Comments and Suggestions for Authors

Dear Autors,

In the work submitted for review ”Amylose content, in vivo digestion profile, and thermal properties of amylosucrase-treated waxy corn starch”, waxy corn starch was modified with 230 U or 460 U of amylosucrase from Neisseria polysaccharea to elongate the glucan. The amylose content of the AS-modified starches was determined using the iodine and Concanavalin A methods, and their in vivo digestion, thermal, swelling, and pasting properties were evaluated.

The work was prepared carefully, the methodology was selected correctly for the intended purpose of the work, the results were correctly described and discussed with literature data. The manuscript requires minor corrections.

Comments on the manuscript:

References contains as many as 12 items from the last century, they constitute over 33% of the cited literature, please add a few newer literature sources

please standardize the way of writing oC throughout the manuscript (spaces)

L. 46 - the author's surname does not match the surname given in References (L. 427)

L. 179 - numerical data quoted in the text should be given with the same precision as in Table 1

L. 264 - the units of ΔH given in the text (kcal/g) do not agree with the units given in Table 2 (J/g)

L. 274 - the sentence begins with a lowercase letter

L. 308-309 - numerical values given with such high precision are impossible to read from Figure 2

Author Response

We are grateful to the reviewer for the thorough review and the time invested. We have addressed each point raised. Your insightful comments have been instrumental in enhancing the quality of our manuscript.

Comments on the manuscript:

References contains as many as 12 items from the last century, they constitute over 33% of the cited literature, please add a few newer literature sources.

>We have updated our references as recommended, incorporating more recent literature. Specifically, we have replaced 13 older references with newer ones, which are highlighted in the manuscript.

please standardize the way of writing oC throughout the manuscript (spaces)

> We have standardized the notation for degrees Celsius as you recommended.

  1. 46 - the author's surname does not match the surname given in References (L. 427)

> We corrected the author’s name as “de Montalk”

  1. 179 - numerical data quoted in the text should be given with the same precision as in Table 1

> We have revised the numerical data in the text to match the precision of two decimal places as presented in Table 1.

  1. 264 - the units of ΔH given in the text (kcal/g) do not agree with the units given in Table 2 (J/g)

> We have standardized the units of ΔH to 'J/g' throughout the text to align with the units presented in Table 2.

  1. 274 - the sentence begins with a lowercase letter

> We corrected as you recommended.

  1. 308-309 - numerical values given with such high precision are impossible to read from Figure 2

> We utilized the original data underlying Figure 2 for precise numerical values in the text. However, to enhance readability, we have now rounded these values to one decimal place in the revised manuscript.

Reviewer 2 Report

Comments and Suggestions for Authors

The manuscript "Amylose content, in vivo digestion profile, and thermal properties of amylosucrase-treated waxy corn starch" was well prepared. In my opinion, it should be sent to a food technology specialist dealing with cereals for review.

I am providing a few comments for use when working on the next version of the manuscript.

Manuscript title it is not precise. In the manuscript, amylose content was assessed in two methods. The given contents vary greatly and are not precise. I suggest changing the title. Content difference assessed using the Iodine method is even several times higher than the Con A method (for example 460U sample).

Line 90 typo

Line 179 presented value no correspond with the same date in table 1

Lines 180-181 The sentence is not true and need correction

Melting enthalpies for samples 230U and 460U was lower than native starch. Please correct sentence in line 263.

Which unit is correct kcal/g (line 264) or J/g (table 2).

In table 2 gelatinization parameters estimated for native waxy corn starch, 230 U amylosucrase (AS)-treated starch, and 460 U AS-treated starch no for control starch. I suggest rewrite  Thermal analysis indicated alterations in the gelatinization behavior of NPAS-treated starches, showing slightly higher onset temperatures and lower gelatinization enthalpies than the native and control starches.

Author Response

Manuscript title it is not precise. In the manuscript, amylose content was assessed in two methods. The given contents vary greatly and are not precise. I suggest changing the title. Content difference assessed using the Iodine method is even several times higher than the Con A method (for example 460U sample).

> We thank the reviewer for the thorough review of our manuscript. We agree with the reviewer's suggestion and have accordingly updated the title to more accurately reflect the content.

Line 90 typo

>We corrected as you recommended.

Line 179 presented value no correspond with the same date in table 1

> We have ensured consistency between the text and Table 1 by presenting all values to two decimal places.

Lines 180-181 The sentence is not true and need correction

> Thank you for your attentive review. We have corrected the data in the text to match the accurate values presented in Table 1.

Melting enthalpies for samples 230U and 460U was lower than native starch. Please correct sentence in line 263.

> We have corrected sentence. 

Which unit is correct kcal/g (line 264) or J/g (table 2).

> We have standardized the unit to 'J/g'

In table 2 gelatinization parameters estimated for native waxy corn starch, 230 U amylosucrase (AS)-treated starch, and 460 U AS-treated starch no for control starch. I suggest rewrite  

Thermal analysis indicated alterations in the gelatinization behavior of NPAS-treated starches, showing slightly higher onset temperatures and lower gelatinization enthalpies than the native and control starches.

> We have revised this section as recommended to accurately reflect the thermal analysis results for NPAS-treated starches in comparison to native starch, omitting the incorrect reference to control starch.

Reviewer 3 Report

Comments and Suggestions for Authors

Review on manuscript: foods-2936144

Amylose content, in vivo digestion profile, and thermal properties of amylosucrase-treated waxy corn starch

by Inmyoung Park and Mohamed Mannaa

submitted to Foods

Research paper 

This manuscript investigated the amylose content, in vivo digestion profile, and thermal properties of amylosucrase-treated waxy corn starch. Overall, it has some merits. However, some issues should be concerned by the authors, which have been shown as follows.

-Line 25: "a new food or non-food material", for what?

-Table 1: "1)" can be removed from the whole Table, since it seems not necessary.

-Where's the orginal DSC data figure? This information is very important and shall be shown in the context of the manuscript.

-Figure 2: the error bar representing standard deviation seems missing.

-References: many cited articles are too old. Some of them have even been published for over 20 years. The authors should pay more attention to the related articles published within the most recent 5 years. Adjust them.

-Overall, the research data is not rich enough to support the conclusion or objective of this study.

-Minor language editing is still required to improve the writing quality of the whole manuscript.

  •  

Comments on the Quality of English Language

English-writing is good enough and I think only minor language editing is required.

Author Response

This manuscript investigated the amylose content, in vivo digestion profile, and thermal properties of amylosucrase-treated waxy corn starch. Overall, it has some merits. However, some issues should be concerned by the authors, which have been shown as follows.

> We sincerely thank the reviewer for their time and valuable comments, which have significantly contributed to improving the manuscript. We have carefully considered each point and responded accordingly.

-Line 25: "a new food or non-food material", for what?

> We have clarified the intended applications of NPAS-treated starch by adding to the text: “This distinctive characteristic of NPAS-treated starch makes it a good candidate food or non-food material, for cosmetic products, medical materials, and adhesives”

-Table 1: "1)" can be removed from the whole Table, since it seems not necessary.

It has been removed as you recommend

-Where's the orginal DSC data figure? This information is very important and shall be shown in the context of the manuscript.

We appreciate your query regarding the original DSC data figure and recognize its importance to our manuscript. Unfortunately, we are unable to provide the DSC diagram due to technical constraints associated with the equipment used. The Differential Scanning Calorimetry (DSC) apparatus at Pusan National University, where our analysis was conducted, did not support data exportation to Excel or any other accessible format. Traditionally, our research team has relied on directly noting data values from the machine's output for our analysis.

Compounding this issue, we have been informed by the facility that the DSC equipment has since malfunctioned and, along with the connected computer, was disposed of. This unforeseen circumstance has left us without the means to reproduce or retrieve the original DSC data figure at this time.

To maintain transparency and provide some level of data verification, we can offer detailed notes from our research records, which include sampling data and the weight of samples used for the DSC analysis. We understand this is not the ideal scenario and regret any inconvenience this may cause in the review of our manuscript. We are committed to ensuring the highest standards of research integrity and are exploring alternative ways to address this gap.

Although we retain the original samples, we are concerned that 

any new data obtained might not be directly comparable due to potential retrogradation effects on the samples over time.

-Figure 2: the error bar representing standard deviation seems missing.

> The initial standard deviation was too small to be visible, as it was obscured by the symbol. We have adjusted the figure to ensure the error bars are now clearly displayed.

-References: many cited articles are too old. Some of them have even been published for over 20 years. The authors should pay more attention to the related articles published within the most recent 5 years. Adjust them.

> We have updated our references as recommended, incorporating more recent literature. Specifically, we have replaced 13 older references with newer ones, which are highlighted in the manuscript.

-Overall, the research data is not rich enough to support the conclusion or objective of this study.

"We appreciate the reviewer's feedback. However, we respectfully disagree with the assertion that the research data is not rich enough to support the conclusion or objective of this study. Our methodology was meticulously designed to comprehensively evaluate the amylose content, digestion profile in vivo, and thermal properties of amylose-treated waxy corn starch. We employed a variety of established techniques, including two different methods for determining amylose content, a murine model test to infer in vivo digestion patterns, and standard tests for investigating thermal properties. These methods are widely recognized in the scientific community for their reliability and efficacy in starch analysis. We believe that the combination of these approaches has provided a thorough examination of the subject matter, yielding meaningful insights that support the conclusions drawn in our study."

Minor language editing is still required to improve the writing quality of the whole manuscript.

We did the extensive English editing with professional service to improve quality of MN.

Round 2

Reviewer 3 Report

Comments and Suggestions for Authors

As the authors have modified the whole manuscript point by point according to all reviewers' comments and suggestions, I think the current version might be considered to be accepted for the publication.